# Multiomics with Evolutionary Computation to Identify Molecular and Module Biomarkers for Early Diagnosis and Treatment of Complex Disease

**DOI:** 10.3390/genes16030244

**Published:** 2025-02-20

**Authors:** Han Cheng, Mengyu Liang, Yiwen Gao, Wenshan Zhao, Wei-Feng Guo

**Affiliations:** 1School of Life Sciences, Zhengzhou University, Zhengzhou 450001, China; chenghan@zzu.edu.cn (H.C.); 15994194193@163.com (M.L.); 15093366018@163.com (Y.G.); zhaowsh07@zzu.edu.cn (W.Z.); 2School of Electrical and Information Engineering, Zhengzhou University, Zhengzhou 450001, China

**Keywords:** evolutionary computation, multiomics, biomarkers, complex disease

## Abstract

It is important to identify disease biomarkers (DBs) for early diagnosis and treatment of complex diseases in personalized medicine. However, existing methods integrating intelligence technologies and multiomics to predict key biomarkers are limited by the complex dynamic characteristics of omics data, making it difficult to meet the high-precision requirements for biomarker characterization in large dimensions. This study reviewed current analysis methods of evolutionary computation (EC) by considering the essential characteristics of DB identification problems and the advantages of EC, aiming to explore the complex dynamic characteristics of multiomics. In this study, EC-based biomarker identification strategies were summarized as evolutionary algorithms, swarm intelligence and other EC methods for molecular and module DB identification, respectively. Finally, we pointed out the challenges in current research and future research directions. This study can enrich the application of EC theory and promote interdisciplinary integration between EC and bioinformatics.

## 1. Introduction

In precision medicine of complex diseases, it is helpful to identify disease biomarkers (DBs) for detecting early warning signals in the critical state and offering potential drug targets [1]. Owing to rapid advancements in omics technologies, huge data resources related to complex diseases have become increasingly available, including gene Copy Number Variation (CNV) data, methylation mutation data, transcriptomics, metabolomics, and pharmacogenomics data, among others. This phenomenon has facilitated the development and testing of novel research models for the early diagnosis, prevention, and treatment of diseases [1]. Multiomics data have recently been pervasively subjected to integrated analyses using a series of computational methods, enabling the identification of potential DBs [2]. Currently, the computational methods for biomarker identification can be broadly categorized into three groups: Complex Network-, Machine Learning (ML)-, and Evolutionary Computing (EC)-based methods. DB identification using complex network systems primarily leverages molecular regulatory networks and their disease development-related dynamic models to analyze critical tipping points and key molecules driving state transitions, thus highlighting the characteristics of system state transitions [3]. Notably, current complex network systems do not consider the diverse dynamic characteristics of optimization objectives in biomarker identification problems (including differences among optimization objectives across multiomics data). Consequently, they may not fully characterize the dynamic properties of DBs, ultimately leading to the loss of important information on DBs, which may affect the early diagnosis and prognostic treatment of complex diseases. To address these shortcomings, considering the multi-level heterogeneous omics data of complex diseases and accurately mining the complex dynamic characteristics of vital biomarker information in molecular networks is imperative. On the other hand, DB identification based on ML classifications could help construct candidate biomarker molecules and non-related biomarker sample sets, extract molecular- and disease-related features, and design classifiers to predict biomarker molecules [4]. Although ML classification methods have shown great potential in improving disease diagnosis and treatment efficiency, they have also been associated with model interpretability challenges. Finally, EC-based DB identification methods, from an optimization standpoint, involve a non-convex, high-dimensional, multi-objective discrete optimization of problems to identify individual critical biomarker molecules. Furthermore, EC-based technologies could address some non-convex complex optimization problems that may not be accurately expressed with functions, offering robustness and global search capabilities, which make them suitable alternatives for such problems. Moreover, recent evolutionary algorithms have shown outstanding performance across various types of optimization problems including high-dimensional, multi-objective, and discrete ones. It is also noteworthy that the natural fit between the essential characteristics of individual key molecule identification problems and the benefits of evolutionary algorithms, along with rapid developments in the field of EC, provide an effective approach to solving individual DB identification problems related to optimization.

Based on the above facts, we reviewed current analysis methods of EC and multiomics, aiming to deeply explore the high-dimensional and diverse dynamic characteristics of DBs, which are classified into three categories according to the types of EC: evolutionary algorithms, swarm intelligence, and other EC methods for identifying node and module DBs using biomolecular omics data. Figure 1 provides an overview of our work. In particular, several representative applications of multiomics in various fields including molecular datasets, analysis, and visualization platforms were discussed in Section 2. Furthermore, in Section 2, current methods for inferring molecular interaction networks are categorized into three types: model-based, information theory-based, and machine learning-based methods. Section 3 reviewed the existing EC-based methods and gave main steps for applying EC on omics for identifying DBs. Finally, Section 4 discussed the future directions of EC-based methods for identifying DBs.

## 2. Datasets for Mining DBs

### 2.1. Applications of Multiomics

Biomedical research has recently undergone significant changes owing to rapid developments in multiomics. These mainly include transcriptomics, proteomics, genomics and epigenomics, single-cell omics, spatial transcriptomics, radiomics, metabolomics, and microbiomics. Furthermore, continuous technological innovations have enabled researchers to explore various biological processes at the molecular level, revealing the complex mechanisms underlying various diseases. In addition to being potent tools for basic research, these technologies have also shown broad prospects in clinical applications. The applications mainly involve the cell molecular level, intestinal microbial system, and pathological imaging, and so on. This section reviews the several representative applications of multiomics tools across various fields.

Multiomics technology has proved useful in cancer research and tailored treatments. Cancer, a complex multi-gene disease involving numerous abnormal gene expressions and mutations, could be comprehensively analyzed through multiomics, elucidating its potential molecular mechanisms and therapeutic options. Furthermore, the increasing popularity of precision medicine and the application of sequencing technology have promoted the development of personalized treatments. Analyzing patients’ specific multiomics patterns could enable clinicians to create tailored treatment plans, improving treatment outcomes and overall patient Survival Rates (SRs). For instance, breast cancer samples were recently subjected to RNA-Seq, revealing the specific expression characteristics of HER2+ breast cancer and providing key data support for targeted therapy against the disease [5]. This molecular feature-guided personalized therapeutic approach is gradually emerging as the new standard in cancer treatment. Moreover, the recent emergence of spatial transcriptomics technology has expanded the dimensions of multiomics research. Transcriptomics integrates multiomics with histomorphological analysis, allowing researchers to resolve multiomics distributions in the spatial context of tissues and organs. For example, spatial transcriptomics technology was used to analyze interactions between immune and cancer cells in the Tumor Microenvironment (TME), elucidating how spatial heterogeneity in the TME impacts tumor progression. Overall, transcriptomics is essential in examining intercellular interactions and multiomics patterns within tissue structures, laying the groundwork for future research [6].

Immunology research and immunotherapy have also benefited greatly from multiomics. The immune system, an important barrier against foreign pathogens, is complex and heterogenous, making it extremely challenging to study. However, through single-cell RNA sequencing (scRNA-Seq) technology, researchers can dissect the immune system’s complexity at the single-cell level, potentially revealing the multiomics differences between different immune cell subsets, which, in turn, could elucidate the immune system’s regulatory mechanisms. Notably, scRNA-Seq has been particularly widely applied in immunotherapy development. In a study on CAR-T cell therapy, the researchers analyzed the functional heterogeneity of T cells using scRNA-Seq technology and discovered that multiomics features correlated closely with treatment efficacy. In addition to helping optimize CAR-T cell therapy strategies, these insights could also indicate directions for future immunotherapy developments. Furthermore, the application of RNA-Seq technology in the study of immune checkpoint inhibitors has greatly advanced [7]. Analyzing immune cell multiomics in the TME could help researchers identify molecular markers that may affect immunotherapy efficacy, thus improving treatment plans and increasing patient response rates.

Multiomics technology is also crucial in neuroscience and brain disease research. The brain, one of the human body’s most complex organs, contains numerous different cell types, each exhibiting distinct multiomics features under different physiological and pathological states. Single-cell RNA sequencing has demonstrated potent capabilities for deeply analyzing the multiomics heterogeneity of different cell types in the brain. For instance, in a study involving Alzheimer’s Disease (AD) patients, the researchers examined the neurons in patients’ brain tissue using scRNA-Seq, revealing multiomics changes in specific neuronal subsets closely related to disease progression. In addition to elucidating the pathogenesis of AD, this finding could also provide potential targets for future treatments. Additionally, multiomics technology revealed disease-related gene expression patterns, highlighting its significance in the study of neurological diseases such as schizophrenia and Parkinson’s Disease (PD), as well as its potential to help scientists identify possible causes and therapeutic targets [8]. Other applications such as plasma proteomics for age-related biomarkers in aging research and integrating omics to identify PP2A in kidney disease are provided in Table 1.

### 2.2. Data Resources of Multiomics

With the advancement of multiomics, researchers are able to generate vast amounts of multiomics data, and these resources provide a platform that allows for the storage, sharing, and analysis of these data. These resources support biomarker research efforts of individual patients and drive new discoveries and innovations. Below are several important types of multiomics data resources, which have opened new prospects for scientific research.

#### 2.2.1. Molecular Data Repositories

Molecular databases focus on specific research areas, providing highly customized data and tools that support in-depth studies of specific diseases or biological processes. The National Center for Biotechnology Information (NCBI) Gene Expression Omnibus (GEO) is a widely used repository for molecular research [13]. It aggregates numerous multiomics research datasets from around the globe, encompassing different species, tissues, and experimental conditions. Researchers can download and reanalyze data from the GEO database for novel insights on pertinent concepts. For instance, they could search the database for the expression data of specific diseases or genes, which they could then use for new hypothesis testing or the validation of existing research results. The European Bioinformatics Institute (EBI) ArrayExpress is another valuable repository, offering numerous multiomics experimental datasets. It supports various data formats and has powerful data searching and downloading functions, enabling researchers to conveniently access and utilize multiomics data [14]. Furthermore, Oncomine, a database specifically for cancer multiomics analysis that aggregates extensive tumor multiomics data, could be leveraged to analyze cancer-related genes, explore cancer progression-related multiomics features, and identify potential DBs or therapeutic targets. This platform integrates tumor multiomics data from multiple studies, offering a powerful tool for cross-study data analysis. On the other hand, ImmPort, a database dedicated to immunology research, integrates various immuno-related multiomics data and provides analytical tools, thus supporting the study of immune responses, immune cells, and disease mechanisms. This platform allows researchers to delve into the immune system’s complexity, uncovering key molecular mechanisms in immune responses and providing data support for developing new immunotherapy strategies [15]. In addition to the aforementioned datasets, the National Institutes of Health (NIH), The Cancer Genome Atlas (TCGA), also offers valuable data resources. It integrates extensive genomic data (including multiomics profiles) comprehensively elucidating the molecular characteristics of cancer. Researchers could search the TCGA database for multiomics data related to different cancer types, which they could then use to identify potential therapeutic targets and explore novel treatment strategies. Overall, the TCGA database allows for the systematic analysis of the molecular basis of cancer, greatly propelling cancer research. The EBI’s Expression Atlas, which integrates multiomics data from different species and conditions, also provides an easy-to-use interface to help researchers conveniently find and analyze the expression patterns of specific genes across different tissues and illnesses. It allows researchers to easily compare gene expression differences under various experimental conditions, enabling them to better understand gene functions and pertinent regulatory mechanisms [16].

#### 2.2.2. Analysis and Visualization Platforms

Various platforms are available that offer data storage, as well as powerful analysis and visualization tools, helping researchers to conveniently process and interpret complex multiomics data. For instance, the Massachusetts Institute of Technology’s (MIT) GenePattern provides a suite of tools for multiomics data analysis, including cluster analysis, Gene Set Enrichment Analysis (GSEA), and comparative analysis of multiomics profiles. Moreover, its user interface is simple and friendly, allowing researchers to perform complex data analyses with simple operations, rapidly yielding meaningful results [17]. Similarly, Galaxy, an open-source genomic data analysis platform, provides a wide range of analysis tools and allows researchers to build and customize data analysis workflows. This platform is highly flexible, making it suitable for analyzing various genomic data types, whether for beginners or for in-depth data mining by experienced researchers [18].

In summary, resources for multiomics data greatly facilitate the development of biology and medical research by providing public data repositories, integrative data resources, specialized databases, and analysis and visualization tools. These resources not only provide ways for researchers to obtain and analyze data but also promote academic collaboration and knowledge sharing on a global scale. With the support of these resources, researchers can explore the complexity of multiomics more deeply, and provide new perspectives and strategies for the prevention, diagnosis, and treatment of diseases.

### 2.3. Biomolecular Network Inference of Individual Patients

Biological network reconstruction using multiomics data is the first step for identifying module DBs in complex disease. Current approaches for inferring molecular interaction networks can be categorized into three general groups: model-, information theory-, and ML-based methods. Model-based methods involve accurate model construction to describe relationships among genes based on fitting gene expression profiles. The constructed model could then be used to build biomolecular networks. Such methods can select different models based on data types and network characteristics, making them highly flexible and scalable. The frequently used model-based methods include Boolean networks [19], Bayesian networks [20], differential equations [21], neural networks [22], and Granger causality methods [23]. On the other hand, information theory-based methods, also known as correlation methods, assume that genes of the same group exhibit similar expression patterns during physiological processes, and measure the correlation between them using a standard metric, with a higher correlation value indicating a higher probability of an interaction between the genes. The frequently employed metrics include the Pearson Correlation Coefficient (PCC), Mutual Information (MI), and Conditional Mutual Information (CMI) [24]. Moreover, some improved versions of these metrics have been employed to infer biomolecular networks, including Mutual Information Correlation-Based Relationship Analysis Tool (MICRAT) [24], Principal Component Analysis-Partial Mutual Information (PCA-PMI) [25], and Conditional Mutual Information to Network Inference (CMI2NI) [26]. In a typical workflow, relationships between variables are first estimated to form a correlation matrix. Significant correlations are then determined via hypothesis testing. The significant correlations are then constructed into a network, where nodes and edges represent variables in the dataset and correlations, respectively. Researchers often introduce prior knowledge of molecular networks to better assess the significance conditions of correlation coefficients (i.e., the threshold for generating edges), thus improving the quality of network inference. Frequently used biological networks in evolutionary optimization principle-based methods for identifying disease markers include, but are not limited to, Protein–Protein Interaction (PPI) networks, metabolic networks, and gene regulatory networks. These methods offer the benefit of low computational complexity and can be leveraged to build large networks even with little data or cases involving a small sample size. Nonetheless, they also have limitations, such as the inability to infer directed networks due to the bidirectional nature of correlations. On the other hand, ML-based methods analyze gene expression data using algorithms and data structures. Notably, regression techniques [27], which are ML-based methods, are often highly interpretable and can be used to determine gene regulation direction, allowing for the reconstruction of directed networks. The ML algorithms frequently used in the reconstruction of biomolecular networks include Gradient-Boosted Trees (GBT), the Markov blanket discovery algorithm, and intervention (do calculus) operations, among others [28,29,30]. The MOVE [31] framework, a methodological structure, was previously employed in examining multiomics variational autoencoders, revealing drug–omics associations in Type 2 Diabetes (T2D), thus highlighting the complex interactions such as between metformin and gut microbiota. Similarly, CEFCON [32], another methodological framework, uses Graph Neural Networks (GNNs) with attention mechanisms to infer cell lineage-specific gene regulatory networks from single-cell RNA data, identifying critical cell fate regulators via the control theory. Despite their potentially valuable contributions, these methods primarily focus on associations rather than causation. Furthermore, while some of these methods leverage the control theory to identify key regulators, they often fail to fully elucidate the causal mechanisms underlying biological processes. These shortcomings underscore the need for data-driven approaches that can integrate both causality and control theory to provide effective interventions. Finally, multimodal data network inference methods use an extended framework of single-modal approaches [including Weighted Gene Co-Expression Network Analysis (WGCNA [33]), Gene Network Inference with Ensemble Trees (GENIE3 [34]), and Single-Cell Regulatory Network Inference and Clustering (SCENIC [35])] to reconstruct Gene Regulatory Networks (GRNs). Specifically, they predict gene expression based on Transcription Factor (TF) gene expression, assign TFs to accessible CREs (Cis-Regulatory Elements) using binding motif information, and associate CREs with genomic distance-constrained target genes. Subsequently, candidate scaffolding networks comprising TF-CRE-gene triplets are generated using multimodal GRN inference methods. Different mathematical strategies are often employed to generate the final GRN structure. Some strategies assume linear relationships between TFs, CRE, and genes (including FigR [36] and GRaNIE [37]), while others assume non-linear relationships (including SCENIC+ [38], CellOracle [39], and Dictys [40]).

It is often difficult to fully comprehend phenotypic changes involved in disease heterogeneity when studying population samples based on molecules or pathways alone [41]. Consequently, several methods have been developed to assess sample-level characteristics by inferring sample-specific interaction networks. These methods include network estimation for single samples through linear interpolation, i.e., Linear Interpolation to Obtain Network Estimates for Single Samples (LIONESS)] [42], construction of Sample-Specific Networks (SSNs) [43,44], construction of Cell-Specific Networks (CSNs) [45], and the use of Single-Sample Network Estimation via Entropic Thresholding (SWEET) [46]. The Paired Sample-Specific Network (Paired-SSN) [44] was previously used to construct specific co-expression networks with paired sample data, such as tumor and normal tissues. These data are particularly suitable for cancer research and offer a new direction for personalized oncology treatment. Furthermore, CSNs [46] are often constructed using single-cell RNA sequencing data, capturing cellular heterogeneity and elucidating both the interactions and communication networks between cells in complex biological systems. On the other hand, the Sample-Specific Pearson Correlation Coefficient (SPCC) [47] constructs networks of functionally dysregulated genes specific to samples, revealing the individual-level molecular network dysregulation, which offers a unique application value in disease subtyping and personalized therapy development. Finally, LIONESS [42] constructs sample-specific regulatory networks from population gene expression data via linear interpolation, unveiling individual–specific regulatory relationships.

The aforementioned methods aim to capture the unique characteristics of each sample or cell, more comprehensively elucidating complex diseases. According to the research, conditional or partial sample-specific correlation networks can be generally used to eliminate indirect co-expressions between genes [48,49]. Furthermore, prior biological experimental validated gene/protein interaction networks can generally be used to extract overlapped edges from the original gene co-expression edges, forming a final personalized gene interaction network for the aforementioned methods. Nonetheless, current single-sample gene regulation network construction methods overlook individual patients’ temporal data [50] and their accuracy and stability also need improvement.

## 3. Evolutionary Computation for Identifying DBs

The EC methods have several advantages, such as ease of use without requiring specific function analysis or gradient information and have a natural advantage in mining disease–omics data with complex dynamic characteristics. These methods are divided into two categories based on the type of DBs, i.e., EC for identifying molecule and module DBs. The applications for some representative EC methods are summarized in Table 2.

### 3.1. Evolutionary Computation for Identifying Molecular DBs

Although isolated, molecular DBs are often sensitive, and specific molecules are associated with certain diseases [51]. More recently, omics data have been subjected to EC feature selection analyses for the discovery of potential DBs. Compared to statistical tests, these approaches often use fewer assumptions regarding data distribution. Owing to recent technological advancements, several EC methods have been developed to help identify molecular DBs. These methods can be divided into three groups: evolutionary algorithms, i.e., Genetic Algorithm (GA) and Genetic Programming (GP)]; swarm intelligence, i.e., Particle Swarm Optimization (PSO) and Ant Colony Optimization (ACO)]; and others. Recently, GAs have been widely employed, particularly in identifying molecular DBs, proving effective. According to research, GA, a powerful optimization method, simulates biological evolution [52]. It leverages operators such as duplication, crossover, and mutation, gradually eliminating fewer fit solutions, thus yielding better solutions over generations. Moreover, GAs excel at searching large solution spaces and could avoid local optima via probabilistic search techniques.

On the other hand, GP methods employ a tree-based structure to generate solutions, with selected features (i.e., candidate DBs) represented as leaf nodes in a tree [53]. It is also noteworthy that GP methods can serve as both a search and classification algorithm. Moreover, GP-based classification methods for identifying DBs have recently become more popular. Across various complex optimization problems, GP methods can generate computer programs, solving potential problems. To find a solution for a specific task, GP methods typically commence with a randomized initial population of individuals. As the process advances, following multiple generations, initial solutions are modified using a specific set of genetic operators, guided by the fitness function. It is at this point that GP-based methods for identifying molecular DBs are introduced. The swarm intelligence methods continuously improve candidate solutions based on a specified fitness measure, thus optimizing molecular biomarker identification problems [54]. First, each particle (i.e., candidate solutions) in the swarm (i.e., population) is assigned a position and velocity. Each particle’s position and velocity are then updated using specific formulas that incorporate the particle’s best-known and global best-known positions. Due to its advantages including enhanced computational performance, ease of implementation, convergence speed, and ability to perform globalized searching, PSO often outperforms other optimization algorithms [55]. However, PSO requires a long running time and a selected high number of features. It also loses the diversity of the swarm as the data dimensions grow, resulting in premature convergence and limiting the exploration of many areas within the search space. On the other hand, ACO, another swarm intelligence method, draws inspiration from the foraging behavior of real ants, where they communicate via pheromone trails to find the most efficient paths to food sources [56]. ACO is characterized by its ability to efficiently exploit and explore potential solutions, perform intelligent searching, global optimization, robustness, and positive feedback for handling high-dimensional, noise, irrelevant and redundant datasets. Meanwhile, it requires high computational time and system resources to obtain the optimal solution.

**Table 2 genes-16-00244-t002:** Summary of EC methods to identify DBs.

Methods	Node Biomarker	Module Biomarker
Genetic Algorithm	- Gene expression-based biomarker discovery [52]; - Identifying predictive radiomic DBs [57]; - The selection of microarray DBs [58,59].	- Identifying disease functional modules [60,61]; - Finding active modules [62]; - Identifying directed signaling pathways [63]; - Identify drug targets [64]; - Identifying biomarker for early warning signals [65].
Genetic Programming	- Selecting gene features for classification of high dimensional mass spectrometry data [66]; - Biomarker discovery in proteomics mass spectrometry data [67]; - Multiple biomarker discovery to diagnose diabetic foot [68].	None
Particle Swarm Optimization	- Identifying non-redundant gene markers from microarray data [69]; - Identification of SNP barcode DBs [70]; - Identification of DBs for arsenicosis [54].	- Finding community of brain networks [71]; - Pathway DBs discovery from gene expression data [72].
Ant Colony Optimization	- Classification of depressive disorders [73]; - Predicting crohn’s disease [56]; - Gene selection for microarray data classification [74]; - Identification of major depressive disorder [75].	- Mining functional modules in protein–protein interaction networks [76,77]; - Identifying Protein Complexes [78].

Notably, other methods such as Differential Evolution (DE) offer several advantages including speed, robustness, and applicability to high-dimensional complex optimization problems in DB identification [79]. However, it has difficulties for setting suitable heuristic parameters to achieve a good performance. There are also other EC methods that have been used to identify DBs. For instance, Yaghoobi et al. [80] employed Modified Multi Objective Imperialist Competitive Algorithm and six objective functions based on the classifier performance/structure evaluation to select microRNAs with biomarker potency in ovarian cancer. Furthermore, Han et al. [81] proposed Zoo algorithm for obtaining the best subset of DBs from large-dimensional datasets using nine integrated swarm intelligence-based feature selection algorithms to vote for the selected features, which were refined by the dynamic recursive feature elimination framework. Additionally, Sharma et al. [82] proposed framework called C-HMOSHSSA for gene selection using multi-objective spotted hyena optimizer (MOSHO) and salp swarm algorithm (SSA) on microarray gene expression data, to facilitate its exploration and exploitation capability.

Despite their good performances in discovering efficient DBs, these ML feature selection methods may still not be used to identify reliable DBs with high accuracy and stability (i.e., Pareto optimal solutions). Due to individual disease heterogeneity, most of these approaches may fail to analyze the predictive power of features as groups and may consider them poor DBs individually. This phenomenon underscores the need for approaches that could identify reliable personalized DBs, representing a promising future research direction.

### 3.2. EC for Identifying Module DBs

In EC-based DB identification methods, the detection of complex disease module DBs is considered a high-dimensional, large-scale discrete optimization problem. Specifically, the problem is selecting suitable molecular sets from large-scale molecular networks and optimizing diagnostic and therapeutic performance indicators. While module DBs align with the natural law of disease evolution, EC offers novel opportunities for understanding system-level disease pathogenesis. However, for complex biological networks, the scale of a network is often much larger, implying that finding the global optimal for large-scale edge networks using current EC methods could be challenging. Therefore, identifying accurate module DBs is imperative to design suitable, large-scale optimization algorithms using more advanced and efficient ML feature selection and complex network techniques that can promote early diagnosis and personalized treatment of complex diseases. Module biomarker identification entails finding important modules that can distinguish between different disease states or progression. In this regard, it is noteworthy that EC methods for identifying module DBs could be categorized into three groups based on different frameworks: Evolutionary algorithms (i.e., GA [60,83]), swarm intelligence [(PSO [72]) and ACO [84])], and DE [85], among other methods [86]. 

For example, by integrating the protein-protein interaction, gene expression, and gene knockout data, a hybrid intelligent method, namely HISP (Hybrid Intelligent approach for identifying directed Signaling Pathways),was proposed to determine the direction of signaling flows within a pathway based on integer linear programming and genetic algorithm [63].Wang et al. [87] proposed an improved multi-objective particle swarm optimization algorithm with penalty boundary intersection decomposition mechanism on gene expression datasets to quantify the relevance of each gene in pathway activity inference. Wu et al. [88] developed a computational framework based on Ant Colony Optimization to rank network nodes where the task of ranking nodes is represented as the problem of finding optimal density distributions of "ant colonies" on all nodes of the network. He et al. [85] established a cooperative co-evolution framework integrated with a new node grouping method and local search operators to identify modules using DE. This framework was designed to effectively handle complex networks of medium to large scales. Meanwhile, they addressed the issue of limited resolution by integrating a recursive partitioning scheme into their approach. Panagiotopoulos et al. introduced a novel hybrid ensemble called MEvA-X, for feature selection (FS) and classification, combining a niche-based multiobjective evolutionary algorithm (EA) with the XGBoost classifier. MEvA-X deploys a multiobjective EA to optimize the hyperparameters of the classifier and perform FS, identifying a set of Pareto optimal solutions and optimizing multiple objectives, including classification and model simplicity metrics [86].

Most DB identification methods tend to focus solely on one specific network or module as a biomarker, neglecting the predictive capabilities of other modules with distinct biomarker configurations. Furthermore, identifying multiple modules (i.e., multi-modal DBs) not only forecasts disease occurrence but also furnishes informative and effective therapeutic drug targets for individual patients. Liang et al. [65] developed a novel multi-modal DB concept and presented a new model (MMPDNB) based on a multi-modal optimization mechanism and the Personalized Dynamic Network Biomarker (PDNB) theory. In addition to providing multiple modules of personalized DBs, this model could also unveil their multi-modal properties. Meanwhile, considering the large-scale property of personalized molecular interaction networks, an improved model MMPDNB-RBM was also developed [89]. This model incorporates the Personalized Dynamic Edge-Network Biomarkers (PDENB) theory, a multimodal optimization strategy, and a latent space search scheme, to identify DBs with different configurations of PDENB modules. Notably, MMPDNB and MMPDNB-RBM are model-free, and their scope of application could be extended. However, due to the large-scale nature of the biological molecules network, obtaining the global optimal solution may not be a guarantee. In the future, researchers could use Decision Variable Analysis (DVA) [90] and dimensionality reduction [91], among other Artificial Intelligence (AI) tools, to develop more effective, large-scale multi-modal optimization algorithms for DB identification.

Structural network control principles, which could provide a theoretically accurate description of how a proper set of driver genes can achieve state transition in PGIN, are quite popular in DB identification [92]. However, in large-scale networks with non-linear dynamics, the optimal set of driver nodes is often computationally difficult to find using structural network control principles, with only the approximate efficient solutions being achievable. Furthermore, these methods only focus on controlling the system through a minimum driver-node set—a single objective optimization-based control and overlook prior drug targets for identifying optimal driver genes. Consequently, Liang et al. [64] recently developed a Multiobjective Optimization Network Control Principle (MONCP)-based novel concept of multi-objective optimization to identify personalized DB in the PGIN for personalized drug targeting. Unlike the traditional structural network control principles, MONCP can provide multiple sets of driver nodes for individual patients, as well as solutions with more target drug information. The researchers also converted MONCP into a discrete multi-objective optimization model with constrained large-scale variables, yielding a novel evolutionary constrained multi-objective algorithm LSCV-MCEA, which could address the challenges associated with the aforementioned model. Notably, LSCV-MCEA adopts a multi-tasking framework, with the primary task aimed at optimizing the original Constrained Multi-Objective Problem (CMOP) with two objectives to approach the Pareto Front (PF), and auxiliary tasks aimed at optimizing the Constrained Single-Objective Problem (CSOP) for each of the two objectives. Briefly, the main population is primarily targeted at a global search of the PF, while the auxiliary populations focus on local research to help the main population explore the undeveloped areas of the PFs. In another study aimed at improving the performance of MONCP, a Knowledge-Embedded Multitasking-Constrained Multi-Objective Evolutionary Algorithm (KMCEA) was proposed, which analyzes the key characteristics of the problems and mines pertinent knowledge to design specific strategies [93]. Furthermore, Liang et al. [94] recently used MONCP on a temporal network observability model to identify disease-predictive DBs.

Nonetheless, MONCP’s variables suffer from a large-scale sparse constraint in the decision space, necessitating the development of more effective strategies, i.e., sparse optimization strategies of CMOPs [95], to improve performance in identifying drug targets. Furthermore, considering the multi-modal optimization of MONCP to find more effective solutions in PGIN presents a promising research direction [96]. Moreover, both multimodal and multi-objective strategies could be used to identify module DBs (Figure 2).

### 3.3. Main Steps of EC for Identifying DBs

In this section, we summarized the main steps for applying EC in multiomics to identify key DBs (Figure 3).

Step 1: Problem formulation

Disease biomarker identification is a typical multi-objective optimization problem (MOP). Generally, a MOP can be summarized as follows:$$\begin{array}{l} min\;F(x)=(f_{1}(x),\;f_{2}(x),\ldots f_{m}(x))\\ \left\{\begin{array}{l} x\in \Omega \\ g_{j}(x)\leq 0,\;j=1,\ldots ,k\\ h_{j}(x)=0,\;j=1,\ldots ,l \end{array}\right. \end{array}$$
where Ω is decision space and *x* is a decision binary variable. *F* denotes objective functions to be optimized, and *m* represents the number of objectives. The optimization function needs to be designed according to requirements, such as the number of DBs, the accuracy of disease sample classification, and the number of effective drug targets [89,94]. $g_{i}(x)$ and $h_{i}(x)$ represent inequality and equality constraints which ensure that the selected DBs guarantee some characteristics of disease-specific pathways and functions (e.g., system controllable) [97]. It should be noted that since these objective functions maybe conflicted with each other in different omics, multi-objective optimization aims to find a set of optimal solutions rather than a single optimal solution.

Step 2: Evolutionary computation strategies design

Since there is no certain paradigm for MOP, it is difficult to use traditional mathematical principles to mine the complex dynamic characteristics of DBs. It is necessary to combine the characteristics of multi-level heterogeneous omics data, and to establish molecular multi-objective optimization models by using EC for disease biomarker identification. For complex diseases, because of the high dimension variables and complex dynamics of the multiomics data, it is necessary to identify multiple sets of predictive performance equivalent DBs to meet the multiscale properties of multisource heterogeneous disease related omics data [97]. Having multiple solutions may help to reveal hidden properties or relations of the disease biomarker identification problem under study, e.g., the distribution of the solution set in the problem space. The design of this EC strategy can be divided into steps such as initialization, evaluation (i.e., Evolutionary individual representations and evaluation functions), environment selection, offspring generation, and final decision.


*Evolutionary individual representations*


The first step in solving the EC-based biomarker selection problem is to choose an appropriate data structure to represent the solution (i.e., DBs). Existing research on solution representations can generally be categorized into four types: vector-based structures, tree-based structures, graph-based structures, and matrix-based structures (Figure 3B–E). The most common individual representation is the vector-based structure, which uses integer/real-valued vectors to denote biomarker selection schemes for GA-based biomarker selection methods. Tree-based representations primarily refer to GP, which can simultaneously optimize biomarker relationships and selection. Graph-based representations mainly pertain to swarm intelligence-based (e.g., Ant Colony Optimization) biomarker selection methods. Matrix-based representations are primarily employed in sparse machine learning algorithms. Each representation method has its own advantages and disadvantages, and selecting the most suitable approach depends on the evolutionary computation method used and the specific requirements for solution representation.


*Evolutionary individual evaluation functions*


The design of individual evaluation functions primarily focuses on the objective function quality of evolutionary individuals but may also incorporate other factors such as model interpretability. For various methods, different individual evaluation functions and representations can be used to measure the importance and contribution of DBs.


*Initialization*


In biomarker selection, it is essential to consider the distribution of initial solutions within the search space. Random initialization may fail to yield satisfactory results. Therefore, existing initialization methods primarily focus on two perspectives (i.e., biomarker quantity and quality) to provide better initial solutions. (i) Methods considering biomarker quantity: These approaches employ strategies such as uniform sampling or probability-adjusted sampling. A well-designed initialization algorithm should ensure that initial solutions are more uniformly distributed in terms of the marker quantity objective, thereby enhancing solution diversity. (ii) Methods considering biomarker quality: These techniques leverage metrics such as correlation and conditional entropy to filter out less significant markers during the initialization phase, thereby generating initial solutions with higher classification performance.


*New solution generation*


The generation of offspring solutions in biomarker selection aims to discover more promising new subsets of markers. Improved offspring generation methods typically focus on two aspects: reducing the number of selected DBs and generating offspring based on biomarker importance. (i) Methods to reduce biomarker quantity: These approaches eliminate irrelevant markers through techniques such as evolutionary pattern mining, unsupervised neural networks, and population distribution detection. For example, sparse optimization techniques are applied to reduce the number of selected markers. (ii) Methods based on biomarker importance: These strategies leverage population distribution information to generate better solutions.


*Environmental Selection*


Environmental selection drives population evolution in biomarker selection, primarily relying on Pareto dominance-based and decomposition-based methods. (i) Pareto dominance-based methods: These are typically used for bi-objective or tri-objective marker selection problems. They select marker subsets through non-dominated sorting and density estimation but may suffer from duplicate marker subsets and local optima issues. (ii) Decomposition-based methods: These convert the problem into multiple single-objective subproblems and solve them using weight generation strategies. However, decomposition methods are sensitive to weight settings and are unsuitable for discontinuous Pareto fronts. Some hybrid methods combine Pareto dominance and decomposition to improve algorithm performance. Others incorporate strategies to filter duplicate solutions in the objective space during environmental selection to maintain population diversity.


*Decision making*


Although marker selection algorithms can generate a set of solutions, only one learning model is typically selected in practical applications. To address this, decision making methods aim to select representative subsets from the non-dominated solution set at the end of the training phase for decision-makers. Three decision strategies are commonly used: (i) Objective preference-based methods: these select final solutions based on the priority of different objectives. (ii) Knee point-based methods: these identify trade-off points, i.e., solutions that incur significant losses in at least one objective but achieve minor improvements in others. (iii) Ensemble-based methods: these integrate multiple marker selection schemes to generate more reliable and robust solutions, mitigating overfitting issues and enhancing the credibility of marker selection results.

Step 3: Performance assessment

To give comprehensive comparisons of the evolutionary optimization principles, the F-score and AUC were commonly adopted to evaluate its performance in identifying DBs on omics dataset. The higher the F-score and AUC are, the better the performance will be. The F-score is utilized to evaluate the enrichment performance of predicted DBs in disease tissue (i.e., breast cancer) specific driver genes [98] and cell specific biomarker genes [99], by computing the harmonic mean of precision and recall. The precision is defined as the ratio of correctly predicted cancer tissue-specific biomarker genes to all the predicted DBs, while recall represents the proportion of correctly predicted cancer tissue-specific DBs to the total number of known cancer tissue-specific DBs in each gold standard list. AUC is the value of the predicted anti-cancer DBs based on predicted probability and the true label in the clinical annotated DBs.

The most frequent criteria are that a biomarker should be good at discriminating between groups of sample areas with desirable performance in terms of F-score and AUC for discriminating samples in early stage, but sometimes other properties may be desirable for clinical applications. For example, (1) survival analysis using DBs can be used to assess the risk associated with these markers. The Kaplan–Meier Plotter (http://kmplot.com) is a commonly used website for performing survival analysis. The data on this website is sourced from databases like GEO (https://www.ncbi.nlm.nih.gov/geo/), and TCGA (https://www.cancer.gov/ccg/research/genome-sequencing/tcga), enabling the evaluation of the correlation between gene expression and patient survival rates across more than 30,000 samples from 21 different types of cancer. This allows for the discovery and validation of DBs related to survival. (2) Furthermore, we can also verify whether DBs are associated with certain important functions of diseases through functional pathway enrichment analysis. Gene set enrichment analysis is a statistical method for identifying associations between gene sets and known biological processes, cellular components, and pathways. These tools use annotated information from databases to identify corresponding gene sets (DAVID (https://david.ncifcrf.gov), GSEA (http://gsea-msigdb.org)). (3) The biological significance of DBs can be validated through the analysis of their association with drug interactions and drug sensitivity. Research on the drug–DB interaction and drug sensitivity of DBs is crucial for achieving personalized treatment for cancer patients and advancing the development of precision medicine. On one hand, a commonly used database for drug–DB interactions is the Drug–Gene Interaction Database (DGIdb) [100], which integrates known drug–gene interactions reported in the literature as well as from over 30 other databases such as DrugBank [101]. On the other hand, one of the most frequently used databases related to drug sensitivity is RNAactDrug (http://bio-bigdata.hrbmu.edu.cn/RNAactDrug/). This database explores the correlation between drug sensitivity and RNA molecules at four molecular levels (expression, copy number variation, mutation, and methylation) through a comprehensive analysis of three large pharmacogenomic databases (GDSC (https://www.cancerrxgene.org/), CellMiner (https://discover.nci.nih.gov/cellminer/home.do), and CCLE (https://sites.broadinstitute.org/ccle)). The date of the above websites is accessed on 14 February 2025.

## 4. Future Directions

Despite advancements in current computational methods for mining cancer omics data for biomarker identification, they still face challenges such as insufficient optimization and characterization capabilities. The multidimensional characteristics of DBs could lead to complex biological effects and individualized patterns in the diagnosis and treatment of diseases, making it difficult to accurately predict the DBs and precisely assess the early critical stages of cancer. Based on these insights, the following research directions should be considered in future studies.

(i)The precise analysis of molecular dynamic characteristics is one of the potential focus areas. In addition to identifying optimized models, resolving the heterogeneity of network features in individual samples could also be crucial in constructing predictive DBs for individual critical states. Although some contemporary methods consider sample-specific perturbation networks [43,45,102] and dynamic responses to gene perturbations [39], they are limited to molecular-level relationships and cannot depict multiomics dynamics across different developmental stages and cell types in cancer patients, including multilayered and multidimensional molecular relationships among drugs, target molecules, and diseases. Furthermore, these methods’ computational complexity and performance could vary based on correlation metrics used, and they may also vary in capabilities to capture non-linear relationships and stochasticity in molecular multiomics. Consequently, mining the dynamic information of critical states from high-dimensional, multisource heterogeneous data in individuals could be challenging. To overcome these issues, key research areas include annotations of paired molecular causal relationships from different dimensions such as genomes, transcriptomes, metabolomes, pathological images, and clinical phenotypes. Others include the construction of training databases, the integration of node and edge feature information across different developmental stages and levels, and establishment of personalized drug–target–disease triadic heterogeneous interaction networks that could represent disease progression and malignant transformations.(ii)Information transfer between different tasks or historical tasks in the high-dimensional space of omics data. Studies have shown that EC theories can identify the complex characteristics of high-dimensional omics data. However, the available optimization techniques, such as co-evolution [85], multi-objective optimization [103,104], and multimodal optimization [65] have not been utilized to study the characteristics of biomarker identification problems, which results in the random utilization of high-dimensional omics data, reducing the application capability of current optimization algorithms in high-dimensional multiomics data space. The efficiency of biomarker identification is directly influenced by the effectiveness of knowledge transfer methods. By leveraging knowledge transfer information from related tasks, such as node and edge markers, collaborative optimization strategies can be designed to enhance the utilization of feasible solutions within the high-dimensional search space and improve overall computational efficiency.(iii)Multiobjectives optimization for dynamic characteristics of biomarkers. Most methods focus on single-objective optimization problems (such as minimizing the number of selected DBs), ignoring the differences in optimization objectives across different omics data [44,105,106]. Single-objective optimization problems typically involve a single optimal solution, which can be identified using established mathematical methods. However, during the selection of cancer DBs, it is essential to minimize the number of molecules, and also include maximum usable information of the selected DBs (for example, accuracy of drug combinations targeting molecules), which implies that the cancer biomarker identification is a multi-objective optimization challenge [64]. For multi-objective optimization problems, there is no fixed paradigm, and when molecular networks change over time, the diverse dynamic characteristics complicate the issue further. A critical future research direction lies in developing a molecular multi-objective optimization model that integrates individual heterogeneity. This model should consider factors such as topological interactions of molecular networks, sparse feature characteristics, and multidimensional biomarker information. By incorporating multitask collaborative optimization and network structure control theory, this model aims to identify more effective individual prognostic DBs for early detection and personalized precision treatment.(iv)Information fusion of omics data. Diagnostic DBs for complex diseases such as cancer present a significant challenge due to the inherent complexity of these diseases. Considering that the pathogenesis of cancer involves genomic, transcriptomic, metabolic, and other multilayered molecular changes, biomarker identification is a high-dimensional non-linear problem. Given the high-dimensional, heterogeneous, and complex dynamic characteristics of cancer omics data, it is often difficult to measure early states with biomarker sets identified by a single data type alone. Researchers should integrate multisource heterogeneous data to identify multiple predictive performance DBs that match the multidimensional characteristics of DBs, compensating for lost or incorrect information in a single data type. However, the majority of the available methods focus on the dimensional differences in omics data, involving a few of the biomarker sets derived from a single data type, ignoring the molecular interaction relationships and high-dimensional characteristics of different data types [4,107,108] which decreases the sensitivity and specificity of early diagnostic DBs. A crucial future research direction involves developing multiview learning feature genetic programming algorithms that effectively consider the complex interactions between different data types. These algorithms should be specifically designed to address the challenges of early critical prediction, such as: acquiring a comprehensive understanding of the complex multidimensional characteristics of early critical DBs; improving the sensitivity and specificity of biomarker detection; leveraging object interaction relationships to enhance the accuracy and robustness of predictive models. By addressing these challenges, we can significantly improve the accuracy and reliability of early critical predictions.(v)Designing EC and deep learning hybrid strategies on omics data for DB discovery. Machine learning-based DB discovery algorithms emphasize the computation of features of molecules or diseases. Although they have achieved certain results, most current models lack biological interpretability [109]. Considering the limitations of the above mentioned methods, some researchers have adopted deep neural networks (i.e., graph neural networks) for extracting the patterns and associations in different types of high-dimensional omics data. Although current deep learning has made some progress in drug identification, most of these achievements ignore optimization ability to mine the complex dynamic characteristics of high dimensional omics data during disease progression. Therefore, by taking advantage of EC methods to deal with high-dimensional omics data, it will ultimately provide more efficient options for early diagnoses and treatment by learning the interrelations among multiple objectives (e.g., the druggability and adverse reactions of DBs) and optimizing the parameters of deep neural networks.

## 5. Conclusions

Designing effective methods to identify valid DBs from high-dimensional multiomics data have always been a focus and challenge in both clinical and basic scientific research. From the perspective of identifying key DBs, it is a non-convex high-dimensional multi-objective discrete optimization problem. Evolutionary computation can solve some complex non-convex optimization problems that are difficult to express accurately with functions. It is robust, globally searchable, easy to apply, and converges quickly, making it an appropriate choice for solving such problems. Additionally, after years of development, EC has shown outstanding performance in various types of optimization problems such as high-dimensional, multi-objective, and discrete issues. The natural fit between the intrinsic characteristics of individual key molecule identification problems and the advantages of swarm intelligence optimization algorithms, along with the rapid development in the field of swarm intelligence, provides an effective method for solving the optimization problems of individual key molecule identification. However, current methods have not systematically summarized the complex characteristics such as the diversity of targets in multiomics data. This paper summarizes multiomics databases and utilizes multi-type high-throughput data from different stages of complex disease development. Considering the advantages of EC in mining the intrinsic characteristics of complex systems, combined with large-scale computing technology, this study explores new intelligent optimization techniques. Through the research in this paper, we aimed to provide new evidence of mining high-dimensional omics data using EC theory, explore the potential medical applications of evolutionary intelligent optimization algorithms, and provide corresponding computational tools and theoretical support for screening early DBs of cancer.

## Figures and Tables

**Figure 1 genes-16-00244-f001:**
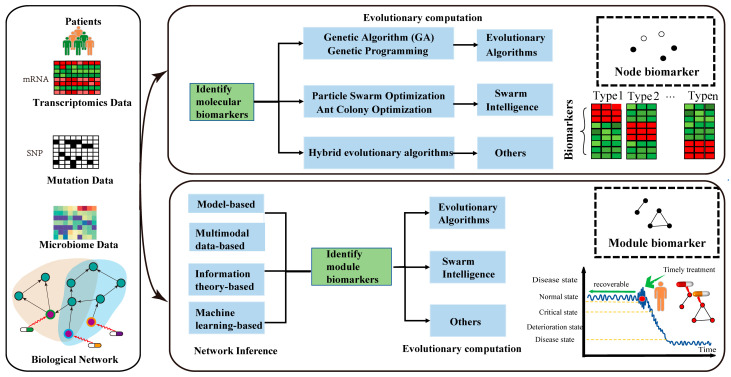
Overview of our review. The contents of our review consists of three parts. Firstly, we summarized several representative datasets of multiomics in various fields. Then, for the multiomics, we pointed out how to identify molecular DBs by using EC tools. Finally, we reviewed the existing methods for inferring biomolecular networks, and EC-based methods for identifying module DBs.

**Figure 2 genes-16-00244-f002:**
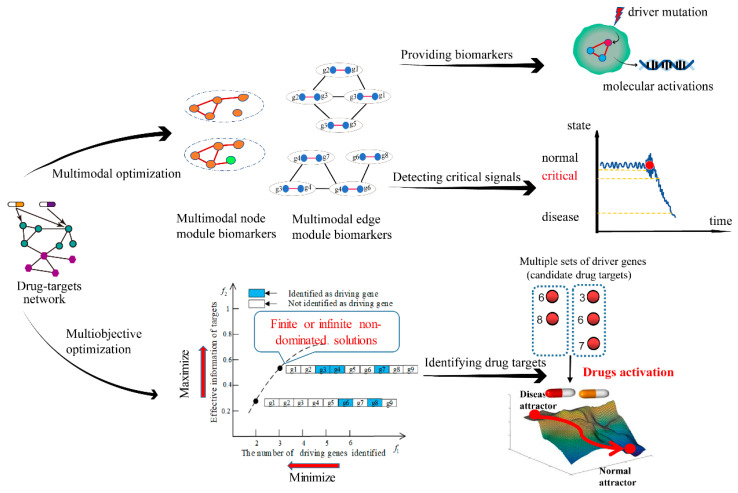
Multimodal and multiobjective strategies for identifying module DBs. On one hand, the multimodal optimization mechanism and personalized dynamic network biomarker theory can provide multiple modules of personalized DBs and unveil their multi-modal properties of node and edge module DBs. On the other hand, multi-objective optimization-based structural network control principles by considering minimum driver nodes and maximum prior-known drug target information can promote early diagnosis and personalized treatment of complex diseases.

**Figure 3 genes-16-00244-f003:**
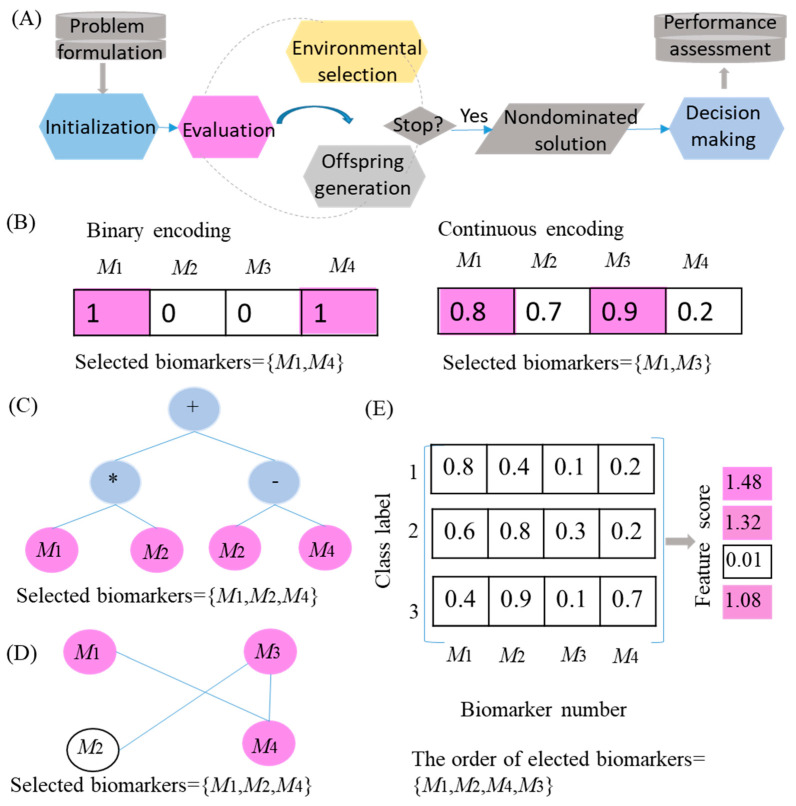
(**A**) Main steps for applying EC in multiomics to identify key DBs. (**B**) Vector-based representation for applying GA-based biomarker selection methods. (**C**) Tree-based representation for applying GP-based biomarker selection methods. (**D**) Graph-based representation for applying swarm intelligence-based biomarker selection methods. (**E**) Matrix-based representations are primarily employed in sparse machine learning algorithms.

**Table 1 genes-16-00244-t001:** The main applications of multiomics.

Application	Description	Examples
DBs and Targets	Identification of molecular markers for diagnosis, prognosis, and therapeutic targets.	- HER2+ breast cancer biomarkers [5] - Spatial transcriptomics in tumor microenvironments [6] - Functional heterogeneity of T cells using scRNA-Seq [7]
Neurodegenerative Diseases	Elucidating molecular mechanisms and identifying therapeutic targets for AD and PD.	- scRNA-seq in AD neuronal heterogeneity [8] - Gut microbiome—metabolome integration in PD [8]
Aging Research	Studying molecular changes linked to aging and age-related diseases.	- Plasma proteomics for age-related DBs [9] - Epigenomic clocks for biological age [10]
Natural Drug Target Discovery	Integrating omics to identify targets of natural compounds for therapeutic development.	- Arctigenin targeting PP2A in kidney disease [11] - Baicalin targeting CPT1 in obesity [12]

## Data Availability

The original contributions presented in the study are included in the article, further inquiries can be directed to the corresponding author.

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
