# Peer review of "Multiomics with Evolutionary Computation to Identify Molecular and Module Biomarkers for Early Diagnosis and Treatment of Complex Disease"

_genes, 2025, doi:10.3390/genes16030244_

Round 1
Reviewer 1 Report
Comments and Suggestions for Authors
Summary
The use of evolutionary computing (EC) for biomarker identification in multiomics research is thoroughly reviewed in this work. Swarm intelligence, evolutionary algorithms, and other EC techniques are the categories into which the authors divide EC-based biomarker identification techniques. They describe the future directions in combining EC with bioinformatics for the development of illness biomarkers and talk about how EC can help with the difficulties of managing high-dimensional, diverse omics data.
Strengths
In computational biology, combining multiomics with EC to identify biomarkers is a new and important area of study.
EC-based techniques are covered in detail in the manuscript, along with an effective classification and an overview of their uses.
From multiomics data to biomarker detection techniques and future prospects, the work flows naturally.
The book presents optimization methods, swarm intelligence, and evolutionary algorithms in a way that even readers who are not familiar with the subject may understand.
Recommendations
Although a variety of EC-based techniques are given, it is unclear how they are used in multiomics.
The description of several EC techniques is dense with text and challenging to understand.
The validation of EC-based biomarker selection in clinical situations is not included in the paper.
Some same concepts are repeated in many parts using different terminology.
There are no specific suggestions for enhancing EC in bioinformatics in the upcoming research section. Talk about how biomarker discovery can be improved by hybrid models (EC + deep learning, for example).
Conclusion
A useful overview of EC-based methods for biomarker detection in multiomics research is given in this paper. To improve clarity and more seriously confront EC's flaws, it has to be revised.

Reviewer 2 Report
Comments and Suggestions for Authors
In this manuscript, titled "Multiomics with evolutionary computation to identify molecular and module biomarkers for early diagnosis and treatment of complex disease”, Cheng and colleagues reviewed current analysis methods of evolutionary computation by considering the essential characteristics of disease biomarker identification problems and the advantages aiming to deeply explore the complex dynamic characteristics of multiomics. The topic is relevant to the field because EC based biomarker identification strategies were summarized as evolutionary algorithms, swarm intelligence and other EC methods for molecular and module DB identification respectively
Compared with other published material authors enrich the application of EC theory and promote interdisciplinary integration between EC and bioinformatics.
The manuscript is overall well written and conclusions are supported by the treated arguments.
Despite thei complexity, treated arguments are clearly presented.
References are appropriate.
Overall figures are well done, although some adjustements are required.
Below my suggestions:
- In the abstract, the explaination of the acronym “EC” should be moved previously;
- In the abstract and in Introduction authors should further highlight the importance of disease biomarker identificationfor the development of personalized medicine;
- Writings in Figure 1 are unfocused;
- Please, add a table summarizing the main applications of multiomics;
- Table 1 is quite confusing. I suggest to schematize the different points and move the relative text in the paragraph;
- In 3.2 please add some concrete examples of biomarkers identification through EC:
- Add a list of abbreviations.
